# Delicate Hybrid Laponite–Cyclic Poly(ethylene glycol) Nanoparticles as a Potential Drug Delivery System

**DOI:** 10.3390/pharmaceutics15071998

**Published:** 2023-07-21

**Authors:** Shengzhuang Tang, Jesse Chen, Jayme Cannon, Mona Chekuri, Mohammad Farazuddin, James R. Baker, Su He Wang

**Affiliations:** 1Michigan Nanotechnology Institute for Medicine and Biological Sciences and Department of Internal Medicine, University of Michigan, Ann Arbor, MI 48109, USA; 2Division of Allergy, Department of Internal Medicine, University of Michigan, Ann Arbor, MI 48109, USA

**Keywords:** functional cyclized polyethylene glycol, PEGylation of laponite, hybrid nanoparticles, drug delivery

## Abstract

The objective of the study was to explore the feasibility of a new drug delivery system using laponite (LAP) and cyclic poly(ethylene glycol) (cPEG). Variously shaped and flexible hybrid nanocrystals were made by both the covalent and physical attachment of chemically homogeneous cyclized PEG to laponite nanodisc plates. The size of the resulting, nearly spherical particles ranged from 1 to 1.5 µm, while PEGylation with linear methoxy poly (ethylene glycol) (mPEG) resulted in fragile sheets of different shapes and sizes. When infused with 10% doxorubicin (DOX), a drug commonly used in the treatment of various cancers, the LAP-cPEG/DOX formulation was transparent and maintained liquid-like homogeneity without delamination, and the drug loading efficiency of the LAP-cPEG nano system was found to be higher than that of the laponite-poly(ethylene glycol) LAP-mPEG system. Furthermore, the LAP-cPEG/DOX formulation showed relative stability in phosphate-buffered saline (PBS) with only 15% of the drug released. However, in the presence of human plasma, about 90% of the drug was released continuously over a period of 24 h for the LAP-cPEG/DOX, while the LAP-mPEG/DOX formulation released 90% of DOX in a 6 h burst. The results of the cell viability assay indicated that the LAP-cPEG/DOX formulation could effectively inhibit the proliferation of A549 lung carcinoma epithelial cells. With the DOX concentration in the range of 1–2 µM in the LAP-cPEG/DOX formulation, enhanced drug effects in both A549 lung carcinoma epithelial cells and primary lung epithelial cells were observed compared to LAP-mPEG/DOX. The unique properties and effects of cPEG nanoparticles provide a potentially better drug delivery system and generate interest for further targeting studies and applications.

## 1. Introduction

Following recent advancements in biomaterials and nanotechnology, drug delivery has undergone enormous developments [1,2,3]. With their flexibility and durability, hybrid organic–inorganic nanomaterials are considered a potential platform with applications in chemistry, physics, life sciences, medicine, and technology [4]. Hybrid nanomaterials based on silicate present an interesting group of materials due to their natural and widely used properties. In the last two decades, laponite (LAP), a synthetic magnesium silicate clay, has emerged as a novel drug delivery nanoplatform [5,6,7]. The dimension of this disc-shaped particle is 25 nm in diameter and 1 nm in thickness with a relatively stable chemical formula of Na^+0.7^[(Mg_5.5_Li_0.3_)Si_8_O_20_(OH)_4_]^−0.7^ [8]. The LAP nanoparticle has a net negatively charged plane and an unstable positively charged edge due to the release of sodium ions on its surface and the protonation of its hydroxyl groups on its edge. Electrostatic interactions amongst the surfaces and edges of the nano discs allows LAP to present as a homogenous dispersion, suspension, or gel, independent of the aqueous system [9,10]. The most basic application of the LAP nanoparticle is a drug–clay hybrid formula that uses the direct mixing of a drug with a LAP aqueous system, and then centrifugation to sediment the composite from solution. The resulting complex may also be further coated with polymer materials for a better release profile.

Chen et al. investigated the absorption of LAP nanoparticles with the enterohemorrhagic E. coli (EHEC) protein and found that the LAP nano-adjuvant was able to induce efficient humoral and cellular immune responses against the EHEC antigen [11]. Kalwar et al. centrifuged an LAP/ciprofloxacin complex, which was then disseminated into polycaprolactone to make nanofibers for more sustained drug release [12]. The silanol group, SiOH, on the edge of the clay sheet creates the potential to chemically modify clay for better solubility and organophilicity. Modifying the edge of the LAP clay using alkoxy silanes possessing additional primary amine groups has been reported by Wheeler et al. [13], allowing more complex polymers to be covalently grafted to the LAP nanoplate, creating a stable hybrid nanomaterial with an inorganic core. For example, a second generation of poly(amidoamine) dendrimer has been conjugated to the LAP nanoplate as a dendrimer-functionalized LAP hybrid nanomaterial [14]. After accreting doxorubicin (DOX), the dispersed composites demonstrated a pH-dependent sustained release profile and more potent inhibitory activities against KB human epithelial cancer cells than free DOX [14].

The excessive accumulation of LAP nano discs might lead to precipitation, so improving the dispersion stability of LAP particles would be a key to enhancing their performance. Polymers are one of the most commonly used stabilizers of inorganic particles such as silicate particles and gold nanoparticles (AuNPs). For example, Ling et al. illustrated different degrees of stability of poly(ethylene glycol) (PEG)-coated AuNPs based on the molecular weight of the linear PEGs used [15]. By adding sodium chloride, the coated AuNPs present visual color changes. Furthermore, the treatment of LAP silicate clay with poly(ethyleneoxide)alkyl ether enhanced stability and resulted in a nanocomposite suspension with spherical particles ranging from 70 nm to 1 µm [16]. Additionally, Gaharwar et al. cross-linked LAP with PEG to make a PEG-silicate nanocomposite hydrogel with flexible interconnective pores, which proved to be mechanically strong and structurally stable while maintaining a high water content [17]. In comparison with linear polymers, cyclized polymers exhibit distinct properties: higher density, higher glass transition temperature, smaller hydrodynamic volume, and lower viscosity [18,19]. By mixing cyclic poly(ethylene glycol) (c-PEG) with AuNPs, Wang et al. proved that physiosorbed c-PEG drastically enhanced the dispersion stability of AuNPs against an external environment and physiological conditions when compared with its linear counterpart [20].

To improve the drug delivery performance, we designed novel hybrid nanoparticles using LAP and cyclic PEG. We first synthesized cyclic PEG with an extra active OH group (cPEG-OH) by which the cyclic PEG was covalently attached to the LAP nanoplate. The chemical homogeneity of synthesized cPEG-OH was confirmed by NMR spectroscopy, specifically ^13^C NMR, mass spectrometry (MS), and gel permeation chromatography (GPC). Moreover, the cPEGylation of LAP to construct the hybrid LAP-cPEG system was characterized by Fourier-transform infrared (FTIR) spectroscopy, ^1^H NMR spectroscopy, dynamic light scattering (DLS), and scanning electron microscopy (SEM). Furthermore, the anticancer drug DOX was captured in the LAP-cPEG system, and the release profile of the LAP-cPEG/DOX formulation was determined in the presence of human plasma. Additionally, the in vitro cytotoxic effect of the LAP-cPEG/DOX formulation was measured by XTT and flow cytometric assays after incubation with A549 lung cancer cells or primary lung epithelial cells.

## 2. Materials and Methods

### 2.1. Materials

LAP (laponite-FN) was provided by BYK Netherlands B.V. (Deventer, Netherlands). Poly (ethylene glycol) 2000 (PEG); methoxy poly (ethylene glycol) 2000 (mPEG); DOX·HCl were obtained from AvaChem Scientific (San Antonio, TX, USA). 4-Nitrophenyl chloroformate; 1,3-diamino-2-propanol (Dimethylamino) pyridine (DMAP); N,N-diisopropylethylamine (DiPEA); 3-aminopropyldimethylethoxysilane (APMES); sodium hydroxide; hydrochloric acid and XTT reagents were all purchased from Sigma-Aldrich (St. Louis, MO, USA). A549 cells were purchased from Japanese Collection of Research Bioresources (JRCB) Cell Bank (Tokyo, Japan). Trypsin-EDTA 0.25%, 7-aminoactinomycin (7-AAD), Fixable Viability Dye eFluor™ 450 were obtained from Thermo Fisher Scientific (Waltham, MA, USA). Alexa Fluor^®^ 647 anti-mouse CD326 (EpCAM) was purchased from BioLegend (San Diego, CA, USA). All solvents were purchased from Sigma-Aldrich and used as received. Deionized (DI) water was used in all the experiments. Dialysis membranes were purchased from Spectrum Laboratories (Rancho Dominquez, CA, USA).

### 2.2. Synthesis of LAP-cPEG Nanoparticles

The fundamental reaction was achieved by preparing the functional cyclic PEG (cPEG-OH). In the current study, cyclized PEG chains were synthesized by a practical and reliable method. First, to make LAP more active, the LAP nano discs were modified with amino groups via a condensation reaction of the LAP’s silanol groups with 3-aminopropyldimethylethoxysilane (APMES) to form LAP-NH_2_ (Figure 1), as described in previous publications [13,14].

Meanwhile, as shown in Figure 2, medium-sized PEG2000 was activated with 4-nitrophenyl chloroformate in the presence of DMAP to form polyethylene glycol dinitrophenyl carbonate (PEG-NP). Purified PEG-NP was treated with equivalent 1,3-diamino-2-propanol to “lock” the terminals of PEG under dilute dichloromethane (DCM) solution in the presence of DiPEA. Thus, a cyclic PEG with a bare hydroxy group was constructed.

Using the same method for the activation of PEG, the desired product (cPEG-OH) was also activated to form cPEG-NP and was then ready for coupling. The active cPEG-NP was treated with LAP-NH_2_ to construct hybrid LAP-cPEG particles (Figure 3).

To be used for comparison, linear mPEG with the same molecular weight as PEG was activated and then coupled to LAP nanoplates using the same procedure as above, forming the LAP-mPEG system (Figure 4).

#### 2.2.1. Procedure for Activation of PEG, mPEG, and cPEG

A single-step method for activation of PEG using 4-nitrophenyl chloroformate (4-NPCl) can produce a series of reactive PEG-phenylcarbonate derivatives. The PEG intermediates are stable for storage, and reaction with amino groups proceeds rapidly at near-neutral pH [21,22].

4-Nitrophenyl chloroformate (3 equiv) in DCM (100 mg/mL) and DMAP (3 equiv) in DCM (100 mg/mL) were added into separate solutions of PEG, mPEG, and cPEG (1 equiv) in DCM (50 mg/mL). The mixtures, now containing PEG-NP, mPEG-NP, and cPEG-NP, were stirred for 16 h at room temperature. After DCM was removed by rotatory evaporation, the residues were triturated from diethyl ether (50 mg/mL, 4 times), re-dissolved in DCM (50 mg/mL), washed with 1 M HCl (50 mg/mL, 2 times), and then with water (50 mg/mL). The resulting solutions were slowly added to an excess volume of diethyl ether (DCM:ether in a 1:10 ratio). The precipitates were filtered and washed with ether and then dried under vacuum.

For production of PEG-NP: activation of PEG (2.0 g, 1.0 mmol) yielded PEG-NP as a white solid (1.4 g, 60%). ^1^H NMR (500 MHz, CDCl_3_): δ 8.28–8.26 (d, J = 10 Hz, 4H, 4ArH-NP), 7.40–7.38 (d, J = 10 Hz, 4H, 4ArH-NP), 4.44–4.42 (t, J = 5 Hz, 4H, 2CH_2_O-(C=O)), 3.82–3.80 (t, J = 5 Hz, 4H, 2CH_2_O-(PEG)), 3.69–3.64 (m, PEG backbone) ppm.

For production of mPEG-NP: activation of mPEG (2.0 g, 1.0 mmol) yielded mPEG-NP as a white solid (1.9 g, 88%). ^1^H NMR (500 MHz, CDCl_3_): δ 8.29–8.27 (d, J = 10 Hz, 2H, 2ArH-NP), 7.40–7.38 (d, J = 10 Hz, 2H, 2ArH-NP), 4.44–4.42 (t, J = 5 Hz, 2H, CH_2_O-(C=O)), 3.82–3.80 (t, J = 5 Hz, 2H, CH_2_O-(PEG)), 3.69–3.54 (m, PEG backbone), 3.37 (s, 3H, CH_3_O) ppm.

For production of cPEG-NP: activation of cPEG (212 mg, 0.1 mmol) yielded PEG-NP as a white solid (186 mg, 81%).^1^H NMR (500 MHz, CDCl_3_): δ 8.28–8.26 (d, J = 10 Hz, 2H, 2ArH-NP), 7.44–7.42 (d, J = 10 Hz,2H, 2ArH-NP), 5.69 (br s, 2H, 2NH), 4.79 (m, 1H, CHO-(C=O)), 4.23–4.21 (t, J = 5 Hz, 4H, 2CH_2_O-(C=O)), 3.78–3.58 (m, PEG backbone) 3.50–3.39 (m, 4H, 2CH_2_N) ppm. ^13^C (500 MHz, CDCl_3_): δ 156.98 (NC=O), 155.45 (OC=O), 151.70 (ArC-NP), 145.40 (ArC-NP), 125.23 (ArC-NP), 121.96 (ArC-NP), 70.50 (PEG backbone), 69.39 (CH_2_O-(C=O)), 64.25 (CH_2_O-(PEG)), 40.04 (CH_2_N) ppm.

#### 2.2.2. Cyclization of PEG-NP to Form cPEG-OH

PEG-NP (300 mg, 0.13 mmol) was dissolved in DCM (300 mL) and cooled in an ice-water bath. A solution of 1,3-diamino-2-propanol (13 mg, 0.14 mmol) in DCM (13 mL) was added dropwise, followed by the addition of DiPEA (136 µL, 0.78 mmol). After being stirred for 24 h at 3–5 °C, the reaction mixture was allowed to warm up to room temperature and was then stirred for another 24 h. The resulting solution was concentrated by rotatory evaporation to 3 mL and slowly added to 30 mL of diethyl ether. The precipitate was filtered and washed with ether, and then further purified by flash column chromatography by eluting with 10% methanol in DCM. The eluted product was triturated from ether, filtered, and dried under vacuum to yield an off-white solid (177 mg, 64%). ^1^H NMR (500 MHz, CDCl_3_): δ 5.66 (br s, 2H, 2NH), 4.22–4.20 (t, J = 5 Hz, 4H, 2CH_2_O-(C=O)), 3.79–3.64 (m, PEG backbone), 3.51–3.49 (m, 1H, CHO), 3.28–3.27 (m, 2H, CH_2_N), 3.19–3.16 (m, 2H, CH_2_N). ^13^C NMR (500 MHz, CDCl_3_): δ 157.29 (C=O), 70.52 (PEG backbone), 70.00 (CHO), 69.48 (CH_2_O-(PEG)), 64.02 (CH_2_O-(C=O), 40.01 (CH_2_N) ppm.

#### 2.2.3. Modification of LAP to Form LAP-NH_2_

LAP powder (100 mg) was suspended in water (80 mL) and stirred while heating at 50 °C overnight for an aqueous dispersion. Then, 32 mL of APMES aqueous solution (2% *w*/*w*, i.e., 1 mL of APMES was combined with 40 mL of water) was added dropwise under vigorous stirring. After stirring at 50 °C for 36 h, the reaction mixture was dialyzed against water (12 times over 3 days) using a dialysis membrane with a molecular weight cut-off (MWCO) of 15,000. The obtained aqueous solution was lyophilized to give a colorless solid LAP-NH_2_ (74 mg).

#### 2.2.4. PEGylation of LAP with mPEG or cPEG to Form LAP-mPEG or LAP-cPEG

First, 60 mg of activated mPEG-NP or cPEG-NP dissolved in acetonitrile (2 mL) was added to 20 mL of an LAP-NH_2_ suspension (1 mg/mL) cooled in an ice-water bath. After stirring for 24 h at 3–5 °C, 1 drop of 1N NaOH was added to maintain the reaction mixture at a pH of 8–9. The mixture was stirred for another 24 h at 3–5 °C and then moved to room temperature and stirred overnight. The clear yellow solution was dialyzed against water (pH 8, 2 times), and water (pH 7.4, 10 times) using a dialysis membrane with an MWCO of 15,000. The final solution was lyophilized to give white solid LAP-mPEG (36 mg) or LAP-cPEG (31 mg).

### 2.3. LAP-PEG/DOX Formulation

LAP-PEG/DOX was formulated by mixing 10% (*w*/*w*) DOX·HCl with LAP-PEG (LAP-mPEG or LAP-cPEG) in water. First, 0.16 mL of DOX·HCl aqueous solution (5 mg/mL) was added to 8 mL of LAP-PEG solution (1 mg/mL). The mixture was stirred for 24 h at room temperature in the dark. The resulting solution was dialyzed against water (150 mL, 4 times) over 36 h in the dark using a membrane with a MWCO of 15,000. The LAP-PEG/DOX (LAP-mPEG/DOX or LAP-cPEG/DOX) formulation in the membrane bag was collected and stored at 3–5 °C in the dark for further application. Meanwhile, the combined dialysis medium was concentrated by rotatory evaporation in reduced pressure and analyzed by UV–Vis with a lambda 25 UV–Vis spectrophotometer (PerkinElmer, Waltham, MA, USA) at 480 nm. The amount of unencapsulated free DOX was determined using a DOX calibration curve. The DOX loading efficiency was determined using the following equation:$$DOX\;loading\;efficiency=\frac{Mass\;of\;feeding\;DOX-Mass\;free\;DOX}{Mass\;of\;feeding\;DOX}\times 100\%$$

### 2.4. In Vitro DOX Release Kinetics from LAP-cPEG/DOX Formulation

The drug release kinetics of the LAP-cPEG/DOX formulation was determined in the presence of PBS or human plasma. Briefly, 1.8 mL of the formulation was mixed well with 0.2 mL of either PBS or human plasma and transferred to a dialysis bag with an MWCO of 15,000. The dialysis bag was then placed in 18 mL of PBS while stirring. At each time interval (0, 0.33, 1, 3, 6, 12, 24 and 48 h), 0.6 mL of PBS buffer was taken for UV analysis and replaced with an equal volume of fresh PBS solution. The concentration of DOX in the dialysis medium was measured using a Lambda 25 UV–Vis spectrophotometer at 480 nm. For comparison, DOX released from the LAP-mPEG/DOX formulation in the presence of plasma was conduct in the same manner.

### 2.5. Antitumor Efficacy of LAP-cPEG/DOX Formulation

#### 2.5.1. XTT Assay

Cell viability was performed via the XTT assay for A549 cells. Cells were planted (15,000 cells/well) in a 96-well tissue culture plate with medium composed of RPMI, 10% FBS, and 1% Pen–Strep, and incubated at 37 °C with 5% CO_2_ overnight. The cells were then fed with DOX, LAP-mPEG/DOX, and LAP-cPEG/DOX to reach the applied DOX concentrations. After 48 h, the supernatants were removed, PBS and XTT reagents were added, and the plate was again incubated for two hours. Using the Synergy HT plate reader (BioTek Instruments; Winooski, VT, USA), the absorbance could be measured as an indication of cell viability, determined from the optical density differences at 690 nm and 492 nm.

#### 2.5.2. Flow Cytometric Assay

A549 cell death was analyzed by 7-AAD staining assay. Primary lung cell death was analyzed by staining with EpCAM (CD326) and Fixable Viability Dye. Briefly, A549 cells (2 × 10^5^) or lung primary cells (1 × 10^6^ cells) were seeded in 48-well plates and incubated at 37 °C with 5% CO_2_ overnight. Concentrated DOX, LAP-mPEG/DOX, or LAP-cPEG/DOX was added to the wells to reach the DOX therapeutic concentration of 0.75 and 1.25 µM for A549 cells or 2 µM for primary lung cells. After 24 h of treatment, cells were trypsinized, washed twice with PBS, stained, and analyzed on the flow cytometer (Novocyte, Agilent, Santa Clara, CA, USA). Flow cytometric data were analyzed using Flow-Jo V10.9 software.

## 3. Results and Discussion

### 3.1. Synthesis of Functional Cyclic PEG

Various approaches to the cyclization of PEG have been attempted [23,24,25]. One of these studies prepared an early version of the fully closed cyclic PEGs as large crown ethers, based on the Williamson reaction in the presence of powdered KOH, a harsh reaction condition [23]. Another common approach uses “click” chemistry, regarded as a mild way to cyclize polymers. A no outlet PEG copolymer was cyclized by click reaction chemistry in the presence of copper cations based on traditional click chemistry [24]. A tadpole-shaped functional copolymer was made by coupling azido-terminated PEG with a dialkyne-terminated polymer [25]. As a result, heavy metals were ultimately introduced to the system due to the formation of stable PEG (Cu^+^) complexes [26,27]. This study provided a gentle approach to synthesize chemically homogeneous cyclic PEG with extra functional groups.

^13^C and ^1^H NMR spectroscopy were used to analyze the cyclic PEG product. As shown in the comparison of the ^13^C spectra of PEG, cPEG-OH, and cPEG-NP (Figure 1a), the signals of neighboring methylene carbons (HO-CH_2_-CH_2_-PEG-CH_2_-CH_2_-OH) of hydroxyl groups at the end of PEG appeared at 72.41 and 61.54 ppm, respectively. After cyclization, the signals moved to 69.48 ppm and 64.02 ppm, respectively. Meanwhile, new single signals appeared at 157.29, 70.00, and 40.01 ppm, belonging to a formed carbonyl carbon and the methylidyne and methylene carbons of 1,3-diamino-2-propanol on cyclized PEG rings, respectively. After activation of cPEG with 4-nitrophenyl chloroformate (4-NPCl) to form cPEG-NP, signals for a new carbamate bond and nitrophenyl group appeared clearly, while the resonance peak of the methylidyne carbon on the cPEG ring shifted downfield due to carbamation. Correspondingly, in the ^1^H spectrum (Figure 1b), after cyclization to form cPEG, the resonance peaks at 3.51–3.49 and 3.28–3.16 ppm representing methylidyne and methylene protons on diamino-propanol were observed, while the resonance peaks of the methylene protons on the terminal PEG backbone shifted downfield due to the formation of the carbamate bond. Nitrophenyl chloroformate activation of cPEG led to a significant downfield shift for the methylidyne proton on the cPEG ring while new resonance peaks of nitrophenyl protons were present after carbamation.

Mass spectrometry (MS) was used to analyze the molecular weight changes in the reaction products. The mass spectra showed an overall increase in molecular weight, centering at 1940.1 for cPEG-OH and 1797.0 for PEG (Figure 2a), in which each peak distribution in cyclic PEG matched well with the precursor by an increase of ca. 142, which corresponds to the molar mass of the monomer lock unit. Incidentally, a regular m/z interval of ca. 44 was observed between neighboring peaks in each distribution for both linear and cyclic PEG, which corresponds to the molar mass of the PEG backbone units.

Gel permeation chromatography (GPC) was used to monitor the changes in linear PEG and cyclic PEG after the reaction. The observed GPC chromatogram of cPEG-OH was unimodal with Mw/Mn = 1.11, compared to mPEG and PEG with Mw/Mn = 1.11 and 1.12, respectively, by the same PEG analytic approach (Figure 2b). While the linear mPEG and PEG had similar retention times, cPEG-OH exhibited a distinctly longer retention time, indicating that the cyclic topology changes the hydrodynamic properties of PEG significantly.

^13^C and ^1^H NMR spectroscopy, mass spectrometry (MS), and gel permeation chromatography (GPC) results indicate cPEG-OH is a homogeneous product without linear PEG mixtures.

### 3.2. PEGylation of Laponite with cPEG and mPEG

The generic route for covalent modification of the LAP surface with additional activated amine groups was previously reported [8,9]. After modification, cPEG-OH or linear mPEG can be covalently attached to LAP nanoplates by the treatment of activated cPEG-NP or mPEG-NP with primary amine-functionalized LAP nanoplates (LAP-NH_2_). As a result of PEGylation, cyclic PEG or linear mPEG molecules were both covalently and non-covalently attached to LAP, respectively. With the minimum loss of LAP during dialysis, there was a 55% weight increase with LAP-cPEG and an 80% weight increase with LAP-mPEG, indicating the composition ratio of LAP:cPEG was about 2:1 for LAP-cPEG, but 2:1.6 for LAP-mPEG.

Fourier transform infrared (FTIR) and ^1^H NMR spectra were used to verify the PEGylation of LAP. As shown in Figure 3a, all the FTIR signals of cPEG-OH were reflected on LAP-cPEG. The peak at 1648 cm^−1^ could be assigned to the carbamate bond formed between cPEG and LAP. In Figure 3b, the PEG backbone has a signal at 3.70 ppm that was observed after PEGylation, and the weaker signals at 3.30 to 3.10 ppm corresponded to the methylene groups of the lock molecule diamino-2-propanol in the LAP-cPEG system. Notably, the signal corresponding to the methylidyne proton in cPEG ring shifted from 3.55 to 4.42 ppm, confirming the formation of the carbamate bond between LAP and cPEG. Moreover, the corresponding integral of the signal ratio is 1:9, indicating that the ratio of covalently attached cPEG: non-covalently attached cPEG is about 1:2 (Appendix A). Similarly, FTIR and NMR studies confirmed the successful coupling of linear mPEG to LAP. An FTIR signal at 1647 cm^−1^ was observed in LAP-mPEG, confirming the formation of the carbonyl bond between mPEG and LAP (Appendix A). In the NMR spectra (Appendix A), an expected peak at 4.65 observed in LAP-mPEG corresponded to the protons adjacent to the terminal hydroxy groups in mPEG while forming the carbamate, and the peak at 3.38 ppm corresponded to the terminal methyl group of mPEG.

### 3.3. Microstructure of PEG-cPEG Particle System

To investigate the size and shape of the synthesized LAP-PEG nano system, a scanning electron microscope (SEM) was used. While LAP particles physically absorbed PEG copolymer surfactant (Brij58) to form rough and stiff spheres ranging from 70 nm to 1000 nm [16] and while cross-linked by PEG copolymer chains led to a tough cellular fiber structure [17], LAP particles attached by cyclic PEG resulted in a distinguishable well-defined structure, a flexible and nearly spherical particle of about 1 µm (Figure 4(a-1,a-2)), in contrast to the attachment of linear PEG, which led to an irregular fiber sheet (Figure 4(b-1,b-2)). cPEGylation of LAP resulted in a stable LAP-PEG system as expected. Higher accumulation of LAP might lead to precipitation from the LAP suspension; however, owing to the specific properties of cPEG, LAP-cPEG remains a clear liquid without precipitate even when stored at lower temperature for longer times (Appendix A). The cPEG endows inorganic LAP particles with greater hydrophilicity and organophilicity.

### 3.4. Dynamic Light Scattering (DLS) Characterization of PEG-cPEG Nanoparticles

The average hydrodynamic size of native LAP varies greatly from 100 nm to 300 nm depending on concentration, which is attributed to the stacking of LAP disc crystals in water [16]. Dynamic light scattering (DLS) was used to monitor the changes in hydrodynamic size and zeta potential of modified LAP (Figure 5). In the study, solid samples of LAP, LAP-NH_2_, and synthesized LAP-PEG were dispersed in water at a concentration of 0.05 mg/mL for DLS analyses. As summarized in Table 1, from LAP to LAP-NH_2_, a slight increase in average size and remarkable increase in average zeta potential were due to the modification of the LAP discs with amine groups, which are positively charged by protonation. The subsequent PEGylation of LAP discs with PEG leads to the expansion of average hydrodynamic size, and the recovery of average zeta potential due to the conversion of amine groups to carbamate groups after coupling. The hydrodynamic size for LAP-cPEG centered at 4661 nm is larger than the size of the mono LAP-cPEG nanoparticle, with an SEM measurement at roughly 1000 nm, indicating the higher order cluster behavior of LAP-cPEG nanoparticles.

### 3.5. Primary Studies of LAP−cPEG/DOX Formulation

#### 3.5.1. LAP−cPEG/DOX Formulation

In this study, 10% DOX was mixed well with LAP-cPEG (1 mg/mL) and then dialyzed against water using a membrane with an MWCO of 15,000 to remove any remaining DOX. The combined dialysates with free DOX were concentrated and quantified by UV–Vis using a standard curve as described in the Appendix A. The drug loading efficiency of LAP-cPEG was found to be 64%, which was higher than that of mPEG (46%). In contrast, when 10% DOX was fed with LAP (1 mg/mL), the LAP/DOX complex precipitated from the solution during dialysis (Appendix A).

The encapsulation of DOX in LAP-cPEG was confirmed by UV–Vis spectroscopy. The characteristic peak of DOX was observed in the absorption spectrum of the LAP-cPEG/DOX formulation with an absorption maximum at around 480 nm (Figure 6a). With its liquid-like properties, the LAP-cPEG/DOX formulation could be characterized by ultra-performance liquid chromatography (UPLC) chromatography. The retention time of the composite slightly lagged in contrast to DOX alone (Figure 6b).

#### 3.5.2. In Vitro Release Studies

The quantitative drug release profile in human plasma was determined by means of dialysis as described in the Appendix A. With the 15 kDa membrane, the free DOX was able to diffuse across the membrane and into the outer medium, which was measured by UV–Vis spectroscopy at predetermined time intervals. As shown in Figure 7a, in the presence of human plasma, compared to the LAP-mPEG/DOX formulation with a burst release of drug at 6 h, the LAP-cPEG/DOX formulation showed a prolonged release profile over 24 h. In the presence of PBS, while almost 100% of DOX was released from DOX solution in less than 3 h, only 15% of the drug was released from the LAP-cPEG/DOX system, indicating that the LAP-cPEG/DOX formulation remained stable in aqueous PBS. Free DOX would form a red precipitate in PBS due to the formation of covalently bonded DOX dimers [28]. Additionally, due to the degradation of free DOX in the presence of plasma [29], a reduction in the concentration of DOX in the measured solution was observed (Figure 7a).

#### 3.5.3. Efficacy of LAP-cPEG/DOX Formulation on A549 Cell Growth Inhibition

After a 48-h incubation, A549 cells had similar levels of viability with DOX or LAP-cPEG/DOX (Figure 7b), suggesting that the DOX in the hybrid nanoparticles can still effectively inhibit cancer cell proliferation. While there was delayed and slow release of DOX from the delivery system, the A549 cell growth inhibition by LAP-cPEG/DOX was found to be similar to DOX in the free form. This indicates that the synthesized LAP-cPEG is functional and has potential as a novel drug delivery system. The OD value in each group was normalized against the OD value in cells cultured with only the appropriate medium.

#### 3.5.4. Increased Drug Efficacy in LAP-cPEG/DOX in Comparison with LAP-mPEG/DOX

At lower DOX concentrations (<0.01 µM), DOX or its LAP-PEG formulations did not affect the survival of A549 cells, while at high DOX doses (e.g., 10 µM), DOX or its LAP-PEG formulations demonstrated similar effects such that almost all A549 cells were killed indiscriminately. In this study, A549 cells were treated with either DOX alone or LAP-PEG/DOX formulations with a final DOX concentration, representing 0.75, 1, and 1.25 µM for 24 h. As summarized in Figure 8, a similar trend was seen in both DOX and LAP-cPEG/DOX treatments, in which cell survival decreased along with increased DOX concentration, whereas cells treated with LAP-mPEG/DOX maintained a consistently higher survival rate. At a DOX concentration of 1 µM or 1.25 µM, LAP-cPEG/DOX was twice as efficient as LAP-mPEG/DOX at inhibiting cancer cell proliferation. Toxicity analysis of these nanoparticles indicated that there was no remarkable interference from the agents (Appendix A). The significant increase in the drug efficacy may be a result of the greater accumulation of LAP-cPEG, which enhanced permeability and retention (EPR) [30,31].

Based on the series of XTT assays in A549 cells, the IC_50_ value for A549 cells was calculated as approximately 1 µM DOX, which corresponded to 9 µg/mL of the LAP-cPEG/DOX complex. This concentration is similar to the free DOX reported in this study. When using this complex to further target specific tumor biomarkers, the advantages of LAP-cPEG/DOX delivery system are enormous due to the lower toxicity.

Flow cytometry was also used to evaluate the effects of the formulations on A549 cells. As shown in Figure 9, LAP-cPEG/DOX kept pace with DOX, displaying rapidly increasing lethality with increasing DOX concentrations from 0.75 to 1.25 µM. In comparison, the effect of LAP-mPEG was much weaker. The 7-AAD staining results were in line with the data from the XTT assay.

#### 3.5.5. Increased Drug Efficacy in LAP-cPEG/DOX in Primary Lung Epithelial Cells

The use of primary lung epithelial cells to evaluate the effects of LAP-cPEG/DOX has the clear advantage of a higher biological relevance compared to A549 cell data. After 24 h incubation, the cells were trypsinized and double-stained with epithelial and live–dead markers for the flow cytometry assay. The result showed that LAP-cPEG/DOX induced apoptosis in about 3% of lung primary epithelial cells, while LAP-mPEG/DOX induced apoptosis in about 1% (Figure 10). This suggests that conjugation of cyclized PEG, but not linear PEG, led to better accumulation and permeation of DOX. Free DOX had similar performance to LAP-cPEG/DOX, consistent with the previous XTT and flow cytometry assays in A549 cells.

## 4. Conclusions

The goal of the current study was to design and assess the feasibility of a new drug delivery system using LAP and cPEG. Chemically homogeneous cyclic PEG with a bare functional hydroxy group was synthesized using gentle conditions. This synthetic approach produces the cPEG in a non-toxic manner and markedly enhances the biocompatibility. The cPEGylation leads to a variety of physical and covalent interactions between LAP nanoparticles and cPEG rings. The increase in surface area enhances the adsorption propensity for organic molecules. Moreover, the presence of cPEG molecules renders the system more hydrophilic and organophilic. LAP-cPEG nanoparticles have a greater solubility and are more biocompatible.

Our drug encapsulation studies indicate that DOX-loaded LAP-cPEG nanoparticles maintain their solubility after being fed with 10% of DOX and have a loading efficiency 1.5 times higher than that obtained with LAP-mPEG. The LAP-cPEG/DOX formulation increased stability over the LAP-mPEG formulation. Moreover, our results have demonstrated that the LAP-cPEG/DOX formulation displays efficient anticancer activity.

The unique properties, release profile, and enhanced cytotoxicity performance encourages further special affinity studies. Furthermore, uniform sizes and morphological factors of the LAP-cPEG nano system would be a priority for its advanced applications. This cPEGylation sets a precedent for constructing these unique organic–inorganic hybrid nanoparticles. Thus, this unique cPEGlyation has further potential in targeting approaches and biological applications.

## Data Availability

Data are contained within the article.

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
