# Peer review of "Delicate Hybrid Laponite–Cyclic Poly(ethylene glycol) Nanoparticles as a Potential Drug Delivery System"

_pharmaceutics, 2023, doi:10.3390/pharmaceutics15071998_

Round 1
Reviewer 1 Report (Previous Reviewer 1)
I appreciate the great efforts that the authors have made in response to my questions and concerns. The revision clarifies all the points I raisedand helps me (and hopefully readers) understand the current manuscript.
Author Response
Reviewer 1:
I appreciate the great efforts that the authors have made in response to my questions and concerns. The revision clarifies all the points I raised and helps me (and hopefully readers) understand the current manuscript.
Response: We are glad, we were able to address reviewer’s questions satisfactory. We appreciate reviewer’s feedback. It helped us improve the manuscript greatly.
Reviewer 2 Report (New Reviewer)
The authors describe the synthesis characterization and drug loading/toxicity of laponite disks modified with cyclic peg for the binding and release of DOX. Overall the paper is written ok with a small need to improve minor english errors etc. Outside of the minor writing errors though there are big gaps in the scientific soundness of this paper resulting in my suggestion in rejection:
Major issues:
The authors describe NPs but all analysis shows them as being micron. Moreover micro nps are nat suitable for the DLS analysis performed as seen by the extremely high PDI (accepted by the field of nps as being outside of the acceptable range).
- The synthesis and characterization of cyclic peg as well as its effects on drug delivery is already well published in the literature. Therfore the first half of the paper is synthesis and characterization that has been similarly performed elsewhere.
-The authors show a release profile without comparing it to the release of the free dox..... this sounds like a wasted control but seeing as how the dox is known to have poor solubility, the control is needed.
-there are 3 different figures all showing the same cytotoxicity repeated (i.e fig 7 and 8) which is then supported by facs Figure 10. While not wasted this means half the paper is based on already published work and the other half based on controllilng toxicity in various ways
- the systems without dox are never represented in cytotoxicity
-see previous point. while the mpeg dox shows lower toxicity no system performs as well or better than the dox alone. so the issue becomes..... wheres the benefit of such an elaborate system. There is abbundant literature showing systems that improve toxicity of doxorubicin in the nano field.
Author Response
We appreciate reviewers feedback. We have answered the raised queries. It helped us improve the manuscript greatly.
Thank you!

Reviewer 3 Report (New Reviewer)
In the present manuscript the authors have designed a new polymeric system aiming to incorporate DXR.
Some comments:
1 1) In the abstract, the authors describe the assays performed. However, at least one sentence, describing the objectives of the present work, is missing.
2 2) In introduction section the sentence that ends in line 40 should include at least one reference.
3 3) In introduction section, the sentence that ends in line 64 – the authors should include which are the tumor cell lin es.
4 4) In the in vitro release studies the authors describe they used 1.8 ml of the formulation with 0,2 ml of either PBS or human plasma. Which was the rationale for such dilution ?
5 5) For the antitumor efficacy of Formulations – section 2.5.1. Did the authors incubate the cells with the formulations at the same time? Please clarify.
6 6) For the 7-AAD staining assay the procedure was different. However for this assay why have been used “concentrated DOX, LAP-mPEG/DOX or LAP-cPEG/DOX”. 24 h after seeding the cells the medium has not been removed? Why the formulations were not solubilized in the complete medium?
7 7) In results section 3.4. a table with the mean size, PdI and zeta potential should be included. The authors refer that these characteristics depend on the concentration used.
8 8) The information regarding the ultra performance liquid chromatography is not included in materials and methods.
9 9) Basically the in vitro studies performed with A549 cells and the 3 formulations under study demonstrated that DXR exhibited similar effect than LAP-cPEG/DOX. Which is the benefit of this formulation over free DXR ?
1 10) In Figure 10 why DXR in the free form was not included in the study?
Author Response
We appreciate reviewers feedback. We have answered the raised queries. It helped us improve the manuscript greatly.
Thank you!

Reviewer 4 Report (New Reviewer)
The problem to which the article is devoted is relevant and currently being studied. However, the manuscript has a number of serious shortcomings that must be taken into account.
1. The text contains paragraphs highlighted in yellow. This is probably the remnants of work on the text that should be removed.
2. The introduction part contains only 15 references, on my opinion it is not enough to form an idea about the research area and set a task that will be solved in the study.
3. What can explain the appearance of the peak at 10 h and the decrease in the % released DOX after 10h (Fig.7a, green and yellow lines)? What processes occur in plasma that are not in the buffer? Add the discussion to the text.
4. What is the porosity and pose size in the developed particles? Molecules in what size range can be encapsulated using these particles?
5. The y-axis in Fig7b must be labeled differently. The results presented on it reflect the cytotoxicity. Therefore, toxicity or viability (in %) should be plotted on the y-axis, as is customary in such cases, and not absorbance! Now this Figure looks unfinished, it is impossible to interpret it.
6. The same issue concerns Fig. 8. What is the IC50 for the developed drug delivery system? This information should be added and discussed.
7. The conclusion section begins with a paragraph of literature review, which should be moved to the discussion section.
Please leave only the results related to the work itself to increase the readability of the manuscript.
8. The text is sloppy from time to time, please make it more readable.
Author Response
We appreciate reviewers feedback. We have answered the raised queries. It helped us improve the manuscript greatly.
Thank you!

Round 2
Reviewer 2 Report (New Reviewer)
The importance of a mild reaction is necessary, but again this reviewer has found numerous other articles producing similar (but not equal constructs) with similar chemistry not using metal catalists or harsh products. Most of my comments still stand even if the authors try to justify them. 25nm particles swelling to more than micro levels. Even with the PDI at 0.5 for micro particles it could be acceptable, so this I will agree. The release profile does help but the others still do not explain what effect the plasma brings to create such a difference in release from pbs (I can immagine there are numerous possibilites) because the effect is drastic with minimal to no release in pbs which seems suspect.
Finally, and most importantly for the value of these systems is the role of these is to have toxicity or specificity. lowering the toxicity compared to dox is not a advantage unless it is shown to increase specific toxicity for the diseased cells which would require coculture or demonstration of less toxicity in non cancerous cells compared to cancerous ones. Otherwise you're limiting the effect of the drug your trying to deliver.
Author Response
Q:The importance of a mild reaction is necessary, but again this reviewer has found numerous other articles producing similar (but not equal constructs) with similar chemistry not using metal catalists or harsh products. Most of my comments still stand even if the authors try to justify them. 25nm particles swelling to more than micro levels. Even with the PDI at 0.5 for micro particles it could be acceptable, so this I will agree. The release profile does help but the others still do not explain what effect the plasma brings to create such a difference in release from pbs (I can immagine there are numerous possibilites) because the effect is drastic with minimal to no release in pbs which seems suspect.
Finally, and most importantly for the value of these systems is the role of these is to have toxicity or specificity. lowering the toxicity compared to dox is not a advantage unless it is shown to increase specific toxicity for the diseased cells which would require coculture or demonstration of less toxicity in non cancerous cells compared to cancerous ones. Otherwise you're limiting the effect of the drug your trying to deliver.
A: Numerous articles reported cyclization of PEG by acetalization or etherification via the tosylate chemistry in the presence of moderate alkalis which results in the fully enclosed cyclic PEG; We herein present an acceptable and concise pathway to produce a cyclic PEG with functional outlet.
In our previous report “Dendrimer-based posaconazole nanoplatform for antifungal therapy”, DRUG DELIVERY 2021, 28, 2150–2159, which showed similar results with the Posaconazole-Dendrimer conjugate drug release profile in plasma and PBS. There are specific components in plasma which trigger the release of the drugs from the delivery system.
We understand that if the toxicity is equally reduced on both cancer cells and normal (non-cancer) cells, there will not be any advantage, or it could become a disadvantage because it dampens the killing effect of DOX on cancer cells. The advantage we described here is that this platform system will reduce the delivery of DOX towards non-target cells/organs, thus decreasing the toxicity towards normal cells/organs. Investigations into the ability targeted delivery of DOX via this platform would be a goal of future research. We thank for the good point you have raised.
We have tried addressing the reviewers’ comments to the best of our ability and knowledge with the given response time. The current manuscript has addressed the advantage of this platform for future targeting delivery systems.

Reviewer 3 Report (New Reviewer)
The authors have replied to all comments.
Some minor points:
The legend in Figure 7 : replace "DOX alone" by "DOX in the free form" .
"IC50" - the "50" should be below the line.
In Table 1 - replace "Particles" by "Nanoparticles"
Author Response
Q: The legend in Figure 7 : replace "DOX alone" by "DOX in the free form" .
"IC50" - the "50" should be below the line.
In Table 1 - replace "Particles" by "Nanoparticles"
A: The legend in Figure 7 “DOX alone” has been replaced by “DOX in the free form”.
“IC50” has replaced by “IC50”.
Table 1 “Particles” has been replaced by “Nanoparticles”
Reviewer 4 Report (New Reviewer)
The authors answered all the questions and improved the quality of the presentation of the results, as well as the text.
Author Response
We appreciate the reviewer’s comments .
This manuscript is a resubmission of an earlier submission. The following is a list of the peer review reports and author responses from that submission.
Round 1
Reviewer 1 Report
In this study, the authors developed Laponite-cyclic poly(ethylene glycol) nanoparticles as a promising drug delivery system. The authors modified laponite nanodisc plates with cyclized poly(ethylene glycol)(PEG) and verified their modification status using FTIR, GPC, and NMR. Then, shape, size, drug loading, and delivery efficiency were evaluated. From these results, the authors concluded that the unique properties and effects of cPEGylation provided a potentially better drug delivery system. The concept of the study is interesting. Despite having a relatively new and important biomedical application, the enthusiasm for the study is tempered for the following reasons.
1. The authors set the goal of better drug delivery through cPeglyzation of Laponite. However, the need for "why" better drug delivery of laponite is questioned. Laponite-based drug delivery systems are mostly long-term delivery (more than 1 month). In this situation, if pegylation is also accompanied, drug delivery efficiency may be significantly reduced. Rather than simply developing new technology, it is necessary to describe what the limitations of LAP nanoparticles are in terms of drug delivery systems and how to solve them with cPEG.
2. A number of studies (mixture/modification both) on PEG/Laponite hybrid hydrogel have already been reported (https://doi.org/10.1039/C0SM00067A, https://doi.org/10.1080/00914037.2016.1182914, https://doi.org/10.1021/acsapm.1c00419). This author's research is unique in that it applied c-PEG modification of laponite particles. What are the advantages of C-PEG modification of laponite particles rather than simply mixing PEG with laponite hydrogel?
3. This reviewer acknowledges that the authors verified the change process through the synthesis process of LAP-cPEG and various methods (NMR, FTIR, mass spectra, gel permeation chromatography). The results supporting what the authors were trying to assert, "LAP-cPEG as a drug delivery system" seems to be quite lacking. In particular, the drug delivery efficiency using LAP-cPEG is only a viability test using one type of cancer cell, and this also does not show a statistically significant difference between groups. The authors should state why the differences between groups are not significant.
4. Although the authors have successfully constructed LAP-cPEG, there is no adequate comparison (controls) in the dox release test and anti-cancer viability. It is recommended to add the LAP/DOX group, which is a negative control group. Also, statistical analysis between groups is missing. Statistical analysis should be added.
5. In Figure 6, when changing from LAP to LAP-cPEG, the size increases considerably (around 500 nm peak to 3000 nm peak), whereas the change in zeta potential is not large. What is the reason for this? LAP forms a "house of cards" structure by electrostatic charge depending on its concentration. Laponite's unique properties act as rheological modifiers that provide shear thinning properties of hydrogel made of Laponite. How do these characteristics change when changing to LAP-cPEG?
6. As the authors stated in the conclusion section, this paper concludes that "LAP-cPEG" is prepared and works in the simplest experiments. With these results alone, it seems difficult to say that a better drug delivery system has been prepared. 1) Changes in properties of c-PEG/LAP according to the degree of modification of c-PEG, 2) The presence or absence of toxicity of c-PEG/LAP itself (LAP induces toxicity and phenotypic changes depending on cells), 3) Anti-cancer study of visualization (live/dead assay, etc.), 4) in vivo efficacy test using cancer model can be added.
Reviewer 2 Report
1.This research studied the drug loading efficiency of the LAP-cPEG nano system and found that it is higher than LAP-mPEG system. The work is popular and has strong novelty, cause the unique properties and effects of cPEGylation provided a potentially better drug delivery system and generate interest for further targeting studies and applications.
2. The experimental design of this research is reasonable, the data is abundant, and the picture processing and explanation are very good, which makes the experimental results can accurately clarify the conclusions.
3.In the article writing aspect, the background and the goal are clear, the experimental method introduction is detailed and has the regulation, discussion analysis is also very clear and understandable.
4. For ease of reading, all images should not be placed on 3.5. According to journal requirements, all figures, schemes and tables should be inserted into the body near their first citation and must be numbered according to their appearance.
5. The synthetic route of LAP-PEG hybrid nanoparticles is clearly shown in figure 1. Since the experimental methods of 2.2.1-2.2.4 have many steps, it is considered that experimental flow chart can be made separately for clear expression.
Round 2
Reviewer 1 Report
I appreciate the authors for their efforts in writing the response letter. Despite their answers, this reviewer is still not convinced in many respects. In particular, the authors described repeated answers (the merits of c-PEG) in the reviewer's comments, but it is difficult to judge that "scientific merit" is clear only with the data they present. In particular, in experiments with cancer cells, it is not considered a valid scientific approach to account for patterns in results that do not have statistical significance.